# Ticagrelor Versus Clopidogrel in Older Patients with NSTE-ACS Using Oral Anticoagulation: A Sub-Analysis of the POPular Age Trial

**DOI:** 10.3390/jcm9103249

**Published:** 2020-10-12

**Authors:** Marieke E. Gimbel, Anne H. Tavenier, Wilbert Bor, Renicus S. Hermanides, Evelyn de Vrey, Ton Heestermans, Melvyn Tjon Joe Gin, Reinier Waalewijn, Sjoerd Hofma, Frank den Hartog, Wouter Jukema, Clemens von Birgelen, Michiel Voskuil, Johannes Kelder, Vera Deneer, Jurriën M. ten Berg

**Affiliations:** 1Department of Cardiology, St. Antonius Hospital, 3435CM Nieuwegein, The Netherlands; w.bor@antoniusziekenhuis.nl (W.B.); keld01@antoniusziekenhuis.nl (J.K.); j.ten.berg@antoniusziekenhuis.nl (J.M.t.B.); 2Department of Cardiology, Isala, 8025AB Zwolle, The Netherlands; r.s.hermanides@isala.nl; 3Department of Cardiology, Meander Medical Centre, 3813TZ Amersfoort, The Netherlands; EA.de.Vrey@meandermc.nl; 4Department of Cardiology, Noord-west Hospital group, 1815JD Alkmaar, The Netherlands; a.a.c.m.heestermans@nwz.nl; 5Department of Cardiology, Rijnstate, 6815AD Arnhem, The Netherlands; mtjon@rijnstate.nl; 6Department of Cardiology, Gelre Hospitals, 7334DZ Apeldoorn, The Netherlands; r.waalewijn@gelre.nl; 7Department of Cardiology, Medical Centre Leeuwarden, 8934AD Leeuwarden, The Netherlands; S.H.Hofma@ZNB.NL; 8Department of Cardiology, Gelderse Vallei Hospital, 6716RP Ede, The Netherlands; hartogf@zgv.nl; 9Department of Cardiology, Leids University Medical Centre, 2333ZA Leiden, The Netherlands; J.W.Jukema@lumc.nl; 10Department of Cardiology, Medisch Spectrum Twente, 7512KZ Enschede, The Netherlands; C.vonBirgelen@mst.nl; 11Department of Cardiology, University Medical Centre Utrecht, 3584CX Utrecht, The Netherlands; mvoskuil@umcutrecht.nl; 12Department of Clinical Pharmacy, Division of Laboratories, Pharmacy, and Biomedical Genetics University, Medical Center Utrecht and Division of Pharmacoepidemiology and Clinical Pharmacology, Utrecht Institute for Pharmaceutical Sciences, Utrecht University, 3584CX Utrecht, The Netherlands; V.H.M.Deneer@umcutrecht.nl

**Keywords:** non-ST-segment elevation myocardial infarction (NSTEMI), elderly, non-vitamin K oral anticoagulation (NOAC), vitamin K antagonists (VKA), ticagrelor, clopidogrel

## Abstract

There are no randomised data on which antiplatelet agent to use in elderly patients with non-ST-elevation acute coronary syndrome (NSTE-ACS) and an indication for oral anticoagulation (OAC). The randomised POPular Age trial, in patients of 70 years or older with NSTE-ACS, showed a reduction in bleeding without increasing thrombotic events in patients using clopidogrel as compared to ticagrelor. In this sub-analysis of the POPular AGE trial, we compare clopidogrel with ticagrelor in patients with a need for oral anticoagulation. The follow-up duration was one year. The primary bleeding outcome was Platelet Inhibition and Patient Outcomes (PLATO) major and minor bleeding. The primary thrombotic outcome consisted of cardiovascular death, myocardial infarction and stroke. The primary net clinical benefit outcome was a composite of all-cause death, myocardial infarction, stroke, and PLATO major and minor bleeding. A total of 184/1011 (18.2%) patients on OAC were included in this subanalysis; 83 were randomized to clopidogrel and 101 to ticagrelor. The primary bleeding outcome was lower in the clopidogrel group (17/83, 20.9%) compared to the ticagrelor group (33/101, 33.5%; *p* = 0.051), as was the thrombotic outcome (7/83, 8.4% vs. 19/101, 19.2%; *p* = 0.035) and the primary net clinical benefit outcome (23/83, 27.7% vs. 49/101, 48.5%; *p* = 0.003). In this subgroup of patients using OAC, clopidogrel reduced PLATO major and minor bleeding compared to ticagrelor without increasing thrombotic risk. This analysis therefore suggests that, in line with the POPular Age trial, clopidogrel is a better option than ticagrelor in NSTE-ACS patients ≥70 years using OAC.

## 1. Introduction

For patients with an acute coronary syndrome (ACS), dual antiplatelet therapy (DAPT) is of utmost importance for the prevention of thrombo-embolic events. The antithrombotic treatment of these patients is more complicated when patients have an indication for oral anticoagulation (OAC), such as atrial fibrillation (AF), because the concomitant use of antiplatelet therapy and oral anticoagulation significantly increases the risk of bleeding [1]. This is especially an issue in elderly patients, as the prevalence of AF rapidly increases with age, and age itself is associated with bleeding [2,3]. European guidelines recommend one to six months of initial triple therapy—a combination of aspirin, clopidogrel and OAC—for patients with ACS and AF who undergo percutaneous coronary intervention (PCI), while the stronger agents ticagrelor and prasugrel are not recommended as part of triple therapy [4]. There are, however, no randomised data on the optimal antiplatelet therapy in this group of patients. Therefore, in this pre-specified subgroup analysis of the POPular AGE trial we compare patients 70 years of age or older with non-ST-elevation ACS (NSTE-ACS) using oral anticoagulation, randomised to clopidogrel or ticagrelor. 

## 2. Materials and Methods

### 2.1. Study Population

The POPular Age trial was an investigator-initiated, randomised, open-label, multicentre study performed in the Netherlands. Previous publications described the methodology of the trial, including study design, in- and exclusion criteria, patient characteristics and outcomes [5,6]. In brief, patients of 70 years or older with NSTE-ACS were randomised 1:1 to ticagrelor/prasugrel or clopidogrel within 72 h of admission. For this sub-analysis, only patients with oral anticoagulation at discharge were included. Until randomisation, patients were treated according to local protocol. After randomisation, patients received a loading dose of the study drug to which they were allocated (clopidogrel 300 or 600 mg, ticagrelor 180 mg or prasugrel 60 mg). Maintenance dosages were used for the duration of one year (clopidogrel 75 mg once daily, ticagrelor 90 mg twice daily, or prasugrel 10 mg once daily). Patients who already used a P2Y_12_ inhibitor at admission and had to be switched according to study protocol received a new loading dose at the discretion of the treating physician. Additional (antithrombotic) treatment was given in accordance to local standards. The trial protocol was approved by a medical research ethics committee and the competent authorities of all study sites. All patients provided written informed consent before enrolment. This trial is registered with the Netherlands Trial Register (NL3804), ClinicalTrials.gov (NCT02317198) and EudraCT (2013–001403–37).

### 2.2. Study Endpoints

The bleeding outcomes were Platelet Inhibition and Patient Outcomes (PLATO) major and minor bleeding at 1 year of follow-up. The composite thrombotic outcome was defined as CV death, myocardial infarction or stroke. The net clinical benefit outcome consisted of all-cause death, myocardial infarction, stroke or PLATO major and minor bleeding. Other outcomes included the abovementioned outcomes at 30-days and the individual components of the combined outcomes at 30 days and 1 year.

### 2.3. Statistical Analysis

Categorical variables were expressed as frequencies and percentages. Comparisons between categorical variables were performed with a Pearson *X^2^* or Fisher exact test in case the proportion of cells with an expected count of <5 exceeded 20%. Continuous variables were presented as mean ± SD or median with interquartile range (IQR), depending on the data distribution. The Student’s *T* test and Mann–Whitney *U* test were used to compare continuous variables, when appropriate. Primary and secondary outcomes were assessed using Kaplan–Meier estimates and statistical differences were tested with the log-rank test. Hazard ratios (HR) with 95% confidence intervals (CI) were estimated by Cox proportional hazard regression, in addition, as per-protocol analysis, to a multivariable analysis including hypertension, diabetes mellitus, history of peripheral arterial disease, history of myocardial infarction, renal function, undergoing coronary angiography, undergoing PCI and undergoing CABG, a sensitivity analysis including patients with OAC at randomisation and a sensitivity analysis including patients who were uneventful during hospital stay and discharged on both OAC and a P2Y_12_ inhibitor. All tests were 2-tailed and used a *p*-value < 0.05 to characterise statistical significance. All analyses were performed with SPSS version 24.0. 

## 3. Results

A total of 1011 patients were enrolled in the POPular Age trial between June 2013 and October 2018. From these, 184 (18.2%) patients used oral anticoagulation in combination with antiplatelet therapy. Of these, 83 (45%) were randomized to clopidogrel and 101 (55%) to ticagrelor/prasugrel. Of the patients in the ticagrelor/prasugrel group, only 1 used prasugrel; therefore, we will refer to the ticagrelor/prasugrel group as the ticagrelor group. The baseline characteristics were described in Table 1. The median age was 77 years; 70% were male, 30% had a prior myocardial infarction and 8% had a previous medical history of ischemic stroke. Before admission, 58.2% used oral anticoagulation and the other patients started oral anticoagulation during their hospital stay. Before randomisation, 56.5% of patients received clopidogrel and 40.8% ticagrelor. At discharge, 69% used VKA (fenprocoumon or acenocoumarol) and 31% NOAC (Table 2). The use of VKA was highest in the first years of recruitment (2013–2015: 87%) and decreased over time (2016–2018: 59%). Furthermore, 27.7% of patients in the clopidogrel group used aspirin and 33.7% in the ticagrelor group. In addition, 22.9% of patients randomised to clopidogrel and 20.8% randomised to ticagrelor were on triple therapy, consisting of aspirin + P2Y_12_ inhibitor + OAC. In total 70% (28/40) of the patients underwent PCI.

During follow-up, premature discontinuation of the study drug occurred in 22 (26.8%) of the patients randomised to clopidogrel (median clopidogrel duration 365 (162–365) days). Of patients randomised to ticagrelor, 32 (31.7%) discontinued, and 42 (41.6%) switched to clopidogrel (median ticagrelor duration 15 (4–195) days; Figure 1A). The most important reason to discontinue clopidogrel was revision of diagnosis (i.e., type II ischemia, no significant stenosis on coronary angiography; 40.9%); for ticagrelor it was the start of oral anticoagulation (35.1%) followed by undergoing CABG during hospital stay (16.2%; Appendix A). A total of 17 (20.5%) patients randomised to clopidogrel discontinued before reaching an endpoint; for patients randomised to ticagrelor this was 58 (57.4%).

The bleeding outcome, consisting of PLATO major and minor bleeding at one year, occurred less often in the clopidogrel group (17/83, 20.9%) compared to the ticagrelor group (33/101, 33.5%; HR 0.55 (95% CI 0.30–1.00), *p* = 0.051, Table 3, Figure 1B), although not significantly. This difference was mainly driven by PLATO major bleeding. The thrombotic outcome of CV death, myocardial infarction and stroke was significantly lower in patients using clopidogrel (7/83, 8.4% vs. 19/101, 19.2%; HR 0.48 (95% CI 0.20–1.14), *p* = 0.035, Table 4, Figure 1C). The net clinical benefit outcome of all-cause death, myocardial infarction, stroke and PLATO major and minor bleeding was significantly lower in patients using clopidogrel compared to ticagrelor (23/83, 27.7% vs. 49/101, 48.5%; HR 0.48 (95% CI 0.29–0.790), *p* = 0.003, Figure 1D). Most events occurred in the first month (Appendix A). The results of the per-protocol analysis were consistent with those of the primary analysis, as were the results of the multivariable analysis, the sensitivity analysis including patients with OAC at randomisation, and the sensitivity analysis including patients with both OAC and a P2Y_12_ inhibitor at discharge without reaching an endpoint during hospital stay (Appendix A).

## 4. Discussion

This sub-analysis of the POPular AGE trial of patients using OAC at discharge showed the following: (1) most (75%) of the patients randomised to ticagrelor had to discontinue ticagrelor or were switched to clopidogrel; (2) in line with the main results of the POPular AGE trial, in this subgroup, clopidogrel reduced PLATO major and minor bleeding compared to ticagrelor, although not significantly; (3) clopidogrel was significantly better in terms of the net clinical benefit outcome compared to ticagrelor.

The majority of patients randomised to ticagrelor were switched to clopidogrel or had to discontinue medication during follow-up. Although the study protocol allowed OAC in combination with ticagrelor, these data show firstly a strong preference for clopidogrel in these elderly patients among treating cardiologist, and secondly a high bleeding rate in the patients randomised to ticagrelor necessitating discontinuation. 

PLATO major and minor bleeding was reduced in patients using clopidogrel with OAC compared to ticagrelor with OAC, which is consistent with the main results of the POPular AGE trial [6]. These results are in line with the RE-DUAL PCI comparing dabigatran with a P2Y_12_ inhibitor versus VKA triple therapy, which showed numerically higher International Society on Thrombosis and Haemostasis (ISTH) major or clinically relevant non-major bleeding rates with ticagrelor (*n* = 327, 21.2% dual therapy with dabigatran, 37.4% triple therapy with VKA) versus clopidogrel (*n* = 2398, 14.5% dual therapy with dabigatran, 25.8% triple therapy with VKA) in both treatment arms [7]. The AUGUSTUS trial also showed similar results, with ISTH major or clinically relevant non-major bleeding in ticagrelor-treated patients being 15.8% (*n* = 52), and 12.2% in clopidogrel-treated patients (*n* = 499) [8].

We also found clopidogrel to be significantly better in terms of the net clinical benefit outcome compared to ticagrelor. Moreover, the thrombotic outcome of CV death, MI and stroke was also significantly lower with clopidogrel. This is in contrast to the PLATO trial, which showed a significant benefit of ticagrelor in reducing ischemic events, even significantly reducing death [9]. A possible explanation for this discrepancy could be the high switching and discontinuation rate in the ticagrelor group, wherein 46.0% discontinued the P2Y_12_ inhibitor, exposing the patient to an increased thrombotic risk. This was confirmed by our sensitivity analysis of patients on both OAC and a P2Y_12_ inhibitor at discharge, wherein the thrombotic endpoint was similar for both treatment groups. However, these positive results of clopidogrel in the elderly are in accordance with the SWEDEHEART registry, including patients of 80 years or older with ACS, which identified a higher risk of death and bleeding in those patients using ticagrelor [10].

A relatively high number of patients was discharged with triple therapy consisting of aspirin, P2Y_12_ inhibitor and oral anticoagulation, while multiple randomised controlled trials investigating the efficacy and safety of dual versus triple therapy showed a significant reduction in major bleeding with dual therapy [7,8,11,12,13]. The AUGUSTUS trial comparing apixaban to VKA and aspirin to placebo in a 2 × 2 factorial design, which showed an important reduction in major and clinically relevant non-major bleeding without increasing thrombotic risk for patients in the placebo group. However, the study was not sufficiently powered to detect differences in thrombotic risk [8]. Therefore, the ESC guideline recommends triple therapy for the shortest time possible for patients with ACS and AF who undergo PCI [4]. 

It is remarkable that most older patients used VKA instead of NOAC in this trial, while the superior safety of the NOAC has been proven in different randomized controlled trials [14,15,16,17]. A recent meta-analysis of NOAC in elderly patients even showed an increased efficacy in elderly patients compared to younger ones [18]. In addition, the overall risk of major bleeding was similar for NOAC and VKA in the elderly, however apixaban and edoxaban showed a significant reduction in elderly patients’ major bleeding risk. 

In this pre-specified subgroup analysis of the POPular AGE trial, we analysed an important and underrepresented population of older patients using OAC with ticagrelor, providing new insights into the risks and benefits of this treatment strategy. However, this sub-group analysis has limitations. First this analysis was not sufficiently powered to find statistical differences between both treatment strategies, therefore these results are hypothesis-generating and should be interpreted with caution. Second, although patients were randomized to either clopidogrel or ticagrelor, the study had an open-label design, which might have influenced treatment decisions after randomisation, especially with regard to switching and discontinuing ticagrelor and the use of triple therapy. Moreover, patients with a very high bleeding risk might be underrepresented in this trial, since the treating physician might not include these patients because of the possibility of randomisation to ticagrelor. Possibly, the disadvantageous profile of ticagrelor would have been more pronounced when these patients would have been included. Lastly, the discontinuation rate of the P2Y_12_ inhibitor in this subgroup was high, especially in the ticagrelor group. In addition, most patients discontinued before reaching an endpoint. Therefore, some patients were undertreated, reducing the risk of bleeding but increasing the thrombotic risk. This might have biased the results. We performed a per-protocol analysis to correct for this bias. The results of the per-protocol analysis were, however, in accordance with the main results of this sub-analysis. 

Future research may further delineate the role of antithrombotic and antiplatelet therapy in the elderly, as this patient population is confronted with high thrombotic and bleeding risks.

## 5. Conclusions

In line with the main results of the POPular AGE trial, in this subgroup of patients using OAC, clopidogrel reduced PLATO major and minor bleeding compared to ticagrelor, without increasing thrombotic risk. Therefore, this subanalysis suggests that clopidogrel is a better option than ticagrelor in NSTE-ACS patients ≥70 years using OAC.

## Figures and Tables

**Figure 1 jcm-09-03249-f001:**
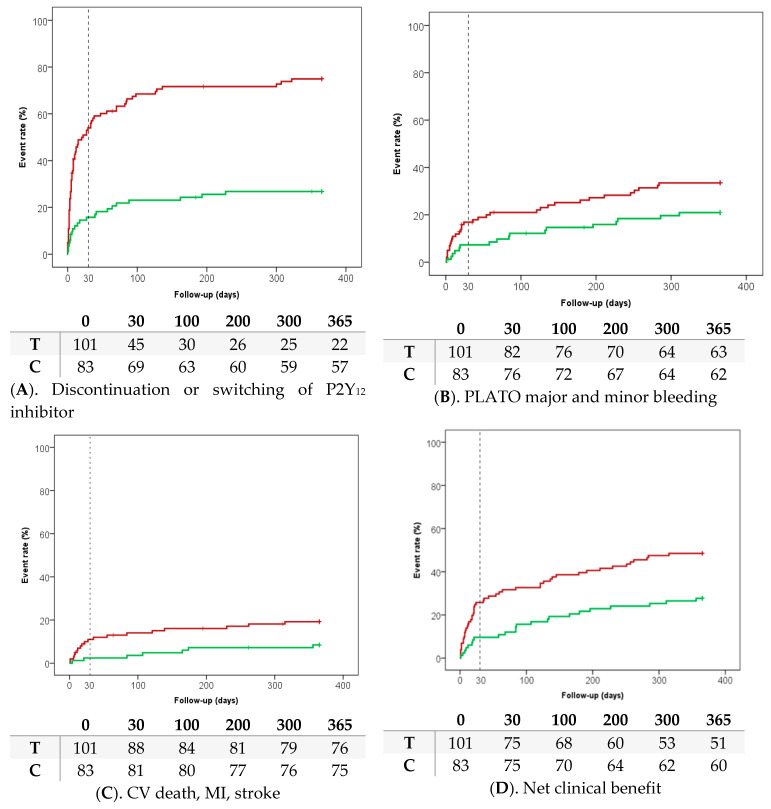
Red patients randomised to ticagrelor (T), green patients randomised to clopidogrel (C). (**A**) Discontinuation or switching of P2Y12 inhibitor; (**B**) PLATO major and minor bleeding; (**C**) CV death, MI, stroke; (**D**) Net clinical benefit.

**Table 1 jcm-09-03249-t001:** Baseline characteristics.

	OAC + Clopidogrel *N* = 83	OAC + Ticagrelor *N* = 101	*p*-Value
Age (median, IQR)	78 (75–83)	77 (73–81)	0.136
Male	58 (69.9)	70 (69.3)	0.933
Body weight < 60 kg	6 (7.2)	7 (7.0)	0.952
**Risk factors**			
Diabetes mellitus	20 (24.1)	42 (41.6)	0.036
Hypertension	65 (79.3)	76 (75.3)	0.534
Hypercholesterolemia	60 (72.3)	65 (63.3)	0.220
Current smoker	7 (8.8)	11 (11.1)	0.443
Family history of CAD	23 (31.1)	23 (25.0)	0.384
**Previous medical history**			
Peripheral artery disease	9 (10.8)	11 (11.0)	0.973
Prior myocardial infarction	29 (34.9)	26 (25.7)	0.175
Prior PCI	21 (25.3)	22 (21.8)	0.575
Prior CABG	19 (22.9)	16 (15.8)	0.225
Transient ischemic attack	7 (8.4)	11 (10.9)	0.577
Ischemic stroke	5 (6.0)	9 (8.9)	0.462
Peptic ulcer	5 (6.0)	4 (4.0)	0.518
COPD	15 (18.1)	9 (8.9)	0.066
**At admission**			
Renal function (median, IQR)	63.7 (48.1–77.4)	62.7 (46.9–79.2)	0.815
eGFR < 60	37 (44.6)	47 (46.5)	0.791
Haemoglobin (median, IQR)	8.7 (8.0–9.3)	8.5 (7.9–9.2)	0.111
Killip class I at admission	74 (90.2)	77 (79.4)	0.050
**During hospital stay**			
Coronary angiography	72 (86.7)	86 (85.1)	0.757
Radial access site	49 (70.0)	56 (69.1)	0.908
Significant coronary lesion	63 (87.5)	78 (90.7)	0.518
multivessel disease	46 (63.9)	57 (66.3)	0.906
Percutaneous coronary intervention	33 (39.8)	36 (35.6)	0.566
CABG	16 (19.3)	28 (27.7)	0.181
**Diagnosis**			
NSTEMI	65 (81.3)	87 (87.0)	0.298
UA	9 (11.3)	5 (5.0)	
Type II ACS	6 (7.5)	8 (8.0)	

Data are % unless stated otherwise. CABG: coronary artery bypass grafting; CAD: coronary artery disease; COPD: chronic obstructive pulmonary disease; eGFR: estimated glomerular filtration rate (CKD-EPI formula); IQR: interquartile range; NSTEMI: non-ST-elevation myocardial infarction; OAC: oral anticoagulation; PCI: percutaneous coronary intervention; SD: standard deviation; triple therapy consists of aspirin + P2Y_12_ inhibitor + OAC; UA: unstable angina.

**Table 2 jcm-09-03249-t002:** Medication at discharge.

	OAC + Clopidogrel *N* = 83	OAC + Ticagrelor *N* = 101	*p*-Value
Aspirin	23 (27.7)	34 (33.7)	0.385
**P2Y_12_ inhibitor**			
Clopidogrel	69 (83.1)	21 (20.8)	<0.001
Ticagrelor	0	57 (56.4)	
Prasugrel	0	1 (1.0)	
no P2Y_12_ inhibitor	14 (16.9)	22 (21.8)	
**Type of OAC**			
VKA	62 (74.7)	65 (64.3)	0.113
NOAC	21 (25.2)	36 (35.7)	
Triple therapy	19 (22.9)	21 (20.8)	0.731
Proton pump inhibitor	73 (88.0)	94 (93.1)	0.233

Data are n (%). NOAC: non-vitamin K oral anticoagulation; OAC: oral anticoagulation; triple therapy consists of aspirin + P2Y_12_ inhibitor + OAC; VKA: vitamin K antagonist.

**Table 3 jcm-09-03249-t003:** Bleeding outcomes at 1 year.

	OAC + Clopidogrel *N* = 83	OAC + Ticagrelor *N* = 101	HR (95% CI)	*p*-Value
PLATO major and minor bleeding	17 (20.9)	33 (33.5)	0.55 (0.30–1.00)	0.051
PLATO minor bleeding	10 (12.4)	14 (14.5)	0.80 (0.35–1.84)	0.614
PLATO other major bleeding	6 (7.4)	7 (7.2)	0.91 (0.30–2.79)	0.971
PLATO major life threatening bleeding	3 (3.7)	12 (12.4)	0.31 (0.09–1.12)	0.038
PLATO major bleeding	8 (9.8)	19 (19.6)	0.47 (0.20–1.09)	0.071
PLATO non-CABG related major bleeding	6 (7.4)	14 (14.5)	0.52 (0.20–1.38)	0.138
ICH	1 (1.2)	2 (2.2)	0.75 (0.07–8.30)	0.645
Fatal bleeding	0	2 (2.1)	-	0.191

Data are n (%). CABG: coronary artery bypass grafting; CI: confidence interval; HR: hazard ratio; ICH: intracranial haemorrhage; OAC: oral anticoagulation; PLATO: Platelet inhibition and patient outcomes.

**Table 4 jcm-09-03249-t004:** Primary net clinical benefit outcome and secondary thrombotic outcomes at 1 year.

	OAC + Clopidogrel *N* = 83	OAC + Ticagrelor *N* = 101	HR (95% CI)	*p*-Value
**Net clinical benefit outcome**				
All-cause death, myocardial infarction, stroke, PLATO major and minor bleeding	23 (27.7)	49 (48.5)	0.48 (0.29–0.79)	0.003
**Second net clinical benefit outcome**				
CV death, myocardial infarction, stroke, PLATO major bleeding	15 (18.1)	35 (35.1)	0.48 (0.26–0.88)	0.008
**Thrombotic outcomes**				
CV death, myocardial infarction, stroke	7 (8.4)	19 (19.2)	0.48 (0.20–1.14)	0.035
All-cause death	4 (4.8)	12 (11.9)	0.42 (0.13–1.32)	0.090
CV death	2 (2.4)	6 (6.0)	0.45 (0.09–2.27)	0.237
Myocardial infarction	5 (6.2)	11 (11.5)	0.60 (0.21–1.74)	0.202
Unstable angina	0	1 (1.1)	-	0.346
Ischemic stroke	1 (1.2)	4 (4.1)	0.30 (0.03–2.77)	0.236
Transient ischemic attack	1 (1.2)	0	-	0.287
Stent thrombosis	0	0	-	-
Urgent revascularisation	1 (1.2)	2 (2.1)	0.74 (0.07–8.18)	0.641

Data are n (%). CI: confidence interval; CV: cardiovascular; HR: hazard ratio; OAC: oral anticoagulation; PLATO; Platelet inhibition and patient outcomes.

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
