# Peer review of "Ticagrelor Versus Clopidogrel in Older Patients with NSTE-ACS Using Oral Anticoagulation: A Sub-Analysis of the POPular Age Trial"

_jcm, 2020, doi:10.3390/jcm9103249_

Round 1

Reviewer 1 Report

This is a sub-analysis on POPular Age trial.

The sample size is really small (n=184), thus data cannot be considered conclusive. To obtain data on bleeding a nd thrombotic event a sample size of more or less 4000-5000 patients is needed ( e.g. AUGUSTUS trial), moreover 2/3 of the patients were on VKA and the remaining in NOAc, thus once again including differencing in the groups, and only 20% of patients were in triple therapy, making data not conclusive.

I have some concerns:

  • Please define also in abstract which is the follow-up time
  • Is POPular Age trial registered somewhere? If so report it please.
  • OAC was performed only with warfarin or also new oral anticoagulants were used? Please specify it in methods.
  • If the trial is randomized, how is it possible that only one patient used prasugrel in the OAC group? Please specify the reason.
  • A p=0.05 cannot be considered statically significant, only a tendency.
  • Please add in Kaplan curves the number of patients and events.
  • Authors declare that this is pre-specified analysis, but there is no mention of this sub-analysis in clinical trial.org (https://clinicaltrials.gov/ct2/show/NCT02317198) and neither in the study protocol (http://dx.doi.org/10.1016/j.ahj.2015.07.030)

Reviewer 2 Report

The article is elegantly written and addresses a pertinent topic of antithrombotic management of high risk patients. Few randomized trials conducted are specific to older people, leading to elderly population being  under-represented in the current evidence base. In this context data presented by Gimbel and colleagues increase in their value as they origin form a sub-analysis of the POPular Age trial, which included elderly (>70 years old) NSTE-ACS patients treated either with ticagrelor/prasugrel or clopidogrel within 72 hours after admission. The current article focuses on a population of patients (18.2% of original POPular Age trial) who used oral anticoagulation in combination with antiplatelet therapy. Those vulnerable patients, along with high risk for major bleeding events, share an increased ischemic burden, as well as the presence of multiple comorbidities adds to the complexity of establishing the optimal treatment strategy. Interestingly, majority of patients randomized to ticagrelor discontinued it or were switched to clopidogrel; clopidogrel was also significantly better in the net clinical benefit outcome compared to ticagrelor, despite a relatively small sample size (184 patients in total). The authors additionally describe in detail the discontinuation pattern and show the dynamics of introduction of NOAC to the daily practice, which increased from the first years of recruitment (2013-2015: 13%) over time (2016-2018: 41%).

As per minor comment, in the main POPular Age trial report authors mentioned that: “Pretreatment with the P2Y12 inhibitor, before coronary angiography, was also at the discretion of the treating physician.” – are the pretreatment data available for the current report?  

Reviewer 3 Report

The authors reported a sub-analysis of the POPular Age trial in the current paper. The original results were published in the Lancet in early 2020, showing that clopidogrel rather than ticagrelor/prasugrel (mostly ticagrelor) was associated with fewer bleeding events without an increase in ischemic events in elderly patients with NSTE-ACS. The present sub-analysis focuses on patients with oral anticoagulation (OAC) at discharge. The authors concluded that the results in this specific population were similar to the original. This reviewer believes that the present paper may provide important data in the field of antithrombotic strategies in coronary artery disease and percutaneous coronary intervention. The biggest issue in this sub-analysis is the low adherence to randomization. At discharge, only 83% and 56% of patients in the clopidogrel and ticagrelor groups were on the assigned study drugs, probably making readers confused. Half of clinical events after discharge in the ticagrelor group was not associated with ticagrelor? The authors may provide another sensitivity analysis that shows the comparison between patients really on OAC plus clopidogrel (n=69) and OAC plus ticagrelor (n=57) at discharge. Basically, guidelines do not recommend the use of ticagrelor and prasugrel as a part of triple therapy (and even as a part of dual therapy), so this reviewer does not understand why so many patients were allowed to receive such unrecommended regimens. The limitations of the present sub-analysis are huge, thus the section should be expanded. Nevertheless, this reviewer does believe that the present data are unique and interesting. The following papers may be useful for the discussion.

  1. Eur Heart J. 2017;38:3070-8.
  2. 2017;13:1168-76.
  3. JAMA Intern Med. 2020;180:420-8.
  4. Cardiovasc Interv Ther. 2020;35:44-51.

Round 2

Reviewer 1 Report

Authors replied to all my questions. The paper is now much clear. I have no other issue to raise.

Reviewer 3 Report

None